# Multifaceted Protective Role of Glucosamine against Osteoarthritis: Review of Its Molecular Mechanisms

**Hiba Murtadha Al-Saadi, Kok-Lun Pang, Soelaiman Ima-Nirwana and Kok-Yong Chin ***

Department of Pharmacology, Faculty of Medicine, Universiti Kebangsaan Malaysia, Cheras,
Kuala Lumpur 56000, Malaysia; hiba.alsaadi2016@yahoo.com (H.M.A.-S.); pangkoklun@gmail.com (K.-L.P.);
imasoel@ppukm.ukm.edu.my (S.I.-N.)
*   Correspondence: chinkokyong@ppukm.ukm.edu.my; Tel.: +603-9145-9573

**Abstract:** Osteoarthritis (OA) is a joint disease resulting from cartilage degeneration and causing joint pain and stiffness. Glucosamine exerts chondroprotective effects and effectively reduces OA pain and stiffness. This review aims to summarise the mechanism of glucosamine in protecting joint health and preventing OA by conducting a literature search on original articles. Current evidence has revealed that glucosamine exhibits anti-inflammatory effects by reducing the levels of pro-inflammatory factors (such as tumour necrosis factor-alpha, interleukin-1, and interleukin-6) and enhancing the synthesis of proteoglycans that retard cartilage degradation and improve joint function. Additionally, glucosamine improves cellular redox status, reduces OA-mediated oxidative damages, scavenges free radicals, upregulates antioxidant proteins and enzyme levels, inhibits the production of reactive oxygen species, and induces autophagy to delay OA pathogenesis. In conclusion, glucosamine prevents OA and maintains joint health by reducing inflammation, improving the redox status, and inducing autophagy in joints. Further studies are warranted to determine the synergistic effect of glucosamine with other anti-inflammatory and/or antioxidative agents on joint health in humans.

**Keywords:** cartilage; chondrocytes; glucosamine; inflammation; osteoarthritis

---

## 1. Introduction

Osteoarthritis (OA) is a common joint disorder affecting millions of people in developed countries. Approximately 13% of women and 10% of men older than 60 years have experienced symptomatic knee OA [1]. The number of people suffering from knee OA may increase due to aging and increasing obesity rate of the general population [2]. OA prevalence is remarkably higher in women than in men and increases with age. OA occurrence at all sites is substantially associated with sex (female) and age (> 50 years) [3]. Strenuous exercise, obesity, joint injury, and genetic susceptibility predispose a person to OA [1,4]. For instance, previous knee trauma increases knee OA risk by 3.86 times [5]. Restriction and modification of risk factors, particularly in weight-bearing joints, may reduce OA risk and prevent subsequent pain and disability [6].

Cartilage erosion and synovial inflammation are some of the morphological changes in OA that are attributed to changes in a complex network of biochemical factors, including proteolytic enzymes that enhance cartilage breakdown [7]. Proinflammatory cytokines, such as interleukin-1 (IL-1) and tumor necrosis factor-alpha (TNF-$\alpha$), are produced by activated synoviocytes, chondrocytes and mononuclear cells, upregulate the expression of matrix metalloproteinases (MMPs) and degrade the cartilage matrix [8]. Various biomarkers, including inflammatory markers (IL-1$\beta$ and TNF-$\alpha$), MMPs, and matrix degradation markers (cartilage oligomeric matrix protein (COMP), C-terminal cross-linking telopeptide of type II collagen (CTX-II), and N-terminal type I collagen telopeptide), can be used to detect OA progression [9–11]. COMP is a 535-kDa non-collagenous protein related to the

thrombospondin family and is primarily found in the articular cartilage, tendons, and synovium [11,12]. CTX-II is a by-product of articular cartilage degradation [13,14]. MMP-3 plays a crucial role in cartilage destruction by degrading the extracellular matrix and activating other MMPs [15,16]. Cytokines, such as TNF-α, IL-1β, IL-6, IL-15, and IL-17, and nitric oxide (NO) are produced by chondrocytes and synoviocytes [17–19]. These molecules exacerbate extracellular matrix degradation; activate additional chondrocytes; and enhance the production of cytokines, aggrecanases, and MMP, leading to chondrocyte apoptosis and cartilage degradation [20,21].

The currently available treatments for OA tend to be focused on symptomatic relief, rather than curative processes, through pain alleviation [22]. For example, non-steroidal anti-inflammatory (NSAIDs) agents are administered to relieve joint pain but they lack efficacy in controlling the disease progression [23,24]. Intra-articular hyaluronic acid injection is used to reduce pain and stiffness of the joints, but its effects are temporary [25,26].

Glucosamine is a type of glycosaminoglycan (GAG), an amino saccharide, and the preferred substrate for the biosynthesis of proteoglycans, such as aggrecans, which maintain cartilage integrity and function [27,28]. Glucosamine reduces proteoglycan loss, delays cartilage degeneration and joint-space narrowing and improves osteoarthritic pain in OA animals and patients [29–32]. In vitro studies showed that glucosamine can reduce the expression of MMPs in chondrocytes and osteoblasts cultures [33,34] and upregulate the expression of collagen type 2A1 and sirtuin-1 (SIRT1) in chondrocytes [34]. Futhermore, glucosamine sulphate (25 mM) or glucosamine hydrochloride (50 mM) completely abrogates calcium ionophore-induced collagen degradation in primary rabbit articular chondrocytes [35]. Glucosamine (175 mg/kg) in combination with chondroitin sulphate (140 mg/kg) reduces joint degeneration in rats with knee OA induced by anterior cruciate ligament transection after 3 months [36]. The positive effects of glucosamine on patients with OA have been established in numerous clinical trials [37–45]. Glucosamine decreases pain, enhances joint function and mobility, and reduces cartilage deterioration in patients with OA [38]. For instance, glucosamine supplementation for 2 years and 6 months has reduced the risk of radiographic knee OA by preventing the narrowing of joint spaces [40]. Long-term supplementation (6 years) of chondroitin sulphate plus glucosamine sulphate has exerted protective effects on joint structures by remarkably reducing cartilage volume loss [45]. Moreover, the combination of ibuprofen or piroxicam (NSAIDs) and glucosamine sulphate for 12 weeks has resulted in lower pain scores than NSAIDs alone on patients with OA regardless of their sex [37]. A randomized, double-blind, placebo-controlled trial was conducted using a mixture of glucosamine hydrochloride (tablet form; 1200 mg/day for 16-week), methylsulfonylmethane, shark cartilage extract, guava leaf extract, 5.6 μg of vitamin D, and 7.35 mg of vitamin B1 on 32 patients with knee OA [46]. This glucosamine mixture significantly reduced joint pain scores, serum cartilage type II collagen degradation and synovitis hyaluronan levels [46]. To date, no adverse effects have been detected in animals and humans treated with glucosamine [39–44].

Glucosamine effectively prevents OA through multiple mechanisms. This narrative review aims to evaluate the current evidence on the mechanism of glucosamine in preventing OA.

## 2. Anti-Inflammatory Activities of Glucosamine

OA is initiated when chondrocytes fail to maintain homeostasis between the synthesis and degradation of extracellular matrix components [47,48]. Inflammation plays a vital role in the early OA development. However, the actual factors initiating the imbalance between cartilage degradation and repair remain elusive [48]. Current findings have suggested that trauma-related microfractures or inflammation in the joint will slightly increase enzymatic activity and 'wear-and-tear' particle formation [48,49]. The overproduction of these particles stimulates phagocytosis by macrophages and the release of degradative enzymes by chondrocytes, thereby promoting the activation and recruitment of macrophages [48,49]. Additionally, the release of proinflammatory cytokines by activated macrophages enhances MMP-mediated cartilage breakdown and leads to synovitis [22].

Considering the role of inflammation in OA, an anti-inflammatory agent should be beneficial in slowing down the progression of this disease [50,51]. Numerous animal [52–55] and human [56] studies reported that glucosamine possesses anti-inflammatory activities in OA. Aghazadeh-Habashi et al. analyzed the effects of different doses of glucosamine (20, 40, 80, or 160 mg/kg/day) on rats with and found that 6-day glucosamine supplementation reduced the levels of proinflammatory cytokines IL-1, IL-6, and TNF-$\alpha$ and prevented the increase in the serum concentrations of nitrite, a stable metabolite of NO in rats with OA [52]. Li et al. reported the effects of 15-day glucosamine treatment (40, 80, and 160 mg/kg; oral) on mice with monosodium iodoacetate-induced OA and found that the mixture of glucosamine and chitooligosaccharides exerts strong anti-inflammatory effects by downregulating serum pro-inflammatory cytokines (IL-1$\beta$, IL-6, and TNF-$\alpha$) and C-reactive protein levels and upregulating the anti-inflammatory cytokine IL-2 level [53]. However, knee joint swelling and typical OA symptoms were partially reduced in the glucosamine-treated groups [53]. A 2-month treatment with 50 and 100 mg/kg glucosamine has remarkably increased anti-inflammatory cytokine IL-10 and reduced transforming growth factor-beta (TGF-$\beta$) levels in monosodium iodoacetate-induced OA in male albino rats [54]. Additionally, male Wistar rats with surgically induced OA were treated with glucosamine sulphate (500 mg/kg) and chondroitin sulphate for 1 month. Similar to the findings of Li et al., these treated OA rats showed substantial improvement in joint histology grading with reduced IL-1$\beta$ expression relative to the negative controls [55].

In a randomized, parallel-group, double-blind clinical trial, elderly patients with temporomandibular joint OA are treated with an intra-articular hyaluronic acid injection or a combination of glucosamine hydrochloride (oral 720 mg for 3 months) and intra-articular hyaluronic acid injection [56]. Both treatments effectively alleviate the symptoms of OA after 1 month of administration, but the combination of glucosamine hydrochloride is greatly beneficial to patients with OA by further reducing the IL-6, IL-1$\beta$, and TGF-$\beta$ levels [56]. Notably, the role of TGF-$\beta$ in OA pathogenesis is not conclusive. TGF-$\beta$ signalling is required for the regulation of chondrocyte homeostasis by favouring chondrocyte proliferation while inhibiting chondrocyte hypertrophy and maturation [57]. Downregulation or inhibition of TGF-$\beta$ signaling is associated with a higher risk of OA incidence [57]. However, Zhen et al. reported contradictory results, wherein the transgenic expression of active TGF-$\beta$ in osteoblasts has induced OA, but the inhibition of TGF-$\beta$ activity in subchondral bone has attenuated the degeneration of articular cartilage [58]. Similarly, Waly et al. revealed that oral glucosamine administration suppresses the early increase in TGF-$\beta$ level in MIA-induced OA rats [54]. However, further study is required to confirm the precise role of TGF-$\beta$ in different stages of OA and the effects of glucosamine on this function.

Mechanistically, the activation of the nuclear factor $\kappa$B (NF-$\kappa$B) and p38 mitogen-activated protein kinase (MAPK) pathway is an integral part of inflammation in OA [59]. Glucosamine sulphate lowers the expression of p38 MAPK and c-Jun N-terminal kinase (JNK) and elevates the extracellular signal-regulated kinase 1/2 (ERK-1/2) expression in the cartilage of rats with OA [32]. Further studies showed that glucosamine prevents cytokine-induced IL-1$\beta$ expression by suppressing the demethylation of the IL-1$\beta$ promoter region [60]. Chiu et al. reported the upstream molecular mechanisms of glucosamine hydrochloride on mouse macrophages J774A.1 cells and human monocytic leukaemia cell line THP-1 [61]. Glucosamine hydrochloride (10 and 30 mM; 2 h treatment) remarkably inhibits lipopolysaccharide (LPS)-induced reactive oxygen species (ROS) generation, NF-$\kappa$B activation, nucleotide-binding oligomerization domain-like receptor containing pyrin domain 3 (NLRP3) inflammasome activation, caspase-1 activation and upregulation and release of IL-1$\beta$ [61]. Moreover, this compound interferes with the binding of double-stranded RNA-activated protein kinase, never-in-mitosis gene A-related kinase 7 and apoptosis-associated speck-like protein containing caspase activation and recruitment domain protein on NLRP3 protein during NLRP3 inflammasome assembly [61].

## 3. Antioxidant Properties of Glucosamine

ROS serves as an important mediator in OA pathogenesis [62] and can be produced during the early OA stage by mechanical stress, trauma, or chemicals. ROS can also trigger cellular damage on the adjacent cartilage, induce collagen degradation, and inhibit proteoglycan synthesis [63]. Antioxidants—such as olive polyphenols, curcumin, and vitamin E—can alleviate pain, restore joint function, and slow down OA progression [17,64–68]. The antioxidant properties of glucosamine have been reported in several in vitro models, including mouse renal mesangial MES-13 cells [69], rat retinal ganglion RGC5 cells [70], macrophage RAW264.7 cells [71], human peripheral lymphocytes [72], and primary human retinal pigment epithelial cells [73]. Glucosamine also reduces oxidative stress in chondrocytes [35,71,74]. In a cell-free system, glucosamine hydrochloride could bind directly with malondialdehyde (MDA) and block the subsequent formation of MDA adducts and protein cross-linkages [74]. Moreover, glucosamine hydrochloride (5 and 50 mM; 4 h co-incubation) inhibits oxidized lipoprotein-induced MDA formation in primary rabbit articular chondrocytes in a concentration-dependent manner [35]. This compound also prevents lipoprotein oxidation and inhibit MDA adduct formation in the articular chondrocyte cell matrix [35]. However, glucosamine hydrochloride does not inhibit the initiation or progression of lipid peroxidation (indicated by conjugated diene formation) either in activated articular chondrocytes or in cell-free copper-induced oxidation of purified lipoproteins condition [35]. Mendis et al. reported that glucosamine hydrochloride and glucosamine sulphate (50–1000 μg/mL) could inhibit hydrogen peroxide ($H_2O_2$)-mediated membrane lipid peroxidation, protein and DNA oxidation in human chondrocytes SW1353 cells in a concentration-dependent manner [71]. Glucosamine sulphate also effectively reduced the intracellular ROS level in chondrocytes SW1353 cells as measured by dichlorodihydrofluorescein diacetate assay [71]. The novel glucosamine-zinc (II) complex [75] and resveratrol-glucosamine hybrid [76] exhibit promising antioxidant activities.

Direct evidence of the antioxidant activity of glucosamine in OA animal models is limited. Currently available in vivo studies were mainly conducted on the rheumatoid arthritis model [75] or in combination with chondroitin sulphate and plant extracts [41]. A combination of glucosamine hydrochloride (175 mg/kg/day), chondroitin sulphate, methylsulfonylmethane, *Harpagophytum procumbens* root extract, and bromelain administered orally for 30 days remarkably reduced MDA, NO, and 8-hydroxyguanine levels and increased total glutathione (GSH) level in formalin-induced osteoarthritic rat knee joint [41]. This combination is more effective than the regimen without plant extracts [41] probably due to the additional analgesic effects [77,78]. Oral intake of glucosamine hydrochloride (300 mg/kg/day for 60 days) has reduced plasma MDA and NO levels in chemical-induced rheumatoid arthritis in albino rats [75]. The combination of glucosamine hydrochloride and alpha-tocopherol acetate (vitamin E; an antioxidant) has further restored the redox status of rheumatoid arthritis in rats [75]. In a clinical trial, patients with knee OA are randomly assigned to two groups of 60 patients, in which the control group receives celecoxib (200 mg/kg/day), and the study group receives glucosamine sulphate (1500 mg/kg/day) in addition to celecoxib for 8 weeks [79]. The glucosamine group has shown better redox status as evidenced by higher serum superoxide dismutase (SOD) activity and lower serum MDA levels compared with the control group [79].

Mechanistically, glucosamine can directly scavenge ROS radicals [71,80–82], upregulate antioxidant enzyme levels and/or activities [71,83], and suppress the endogenous ROS or RNS production by nicotinamide adenine dinucleotide phosphate (NADPH) oxidase and inducible nitric oxides synthase (iNOS) enzymes [84–86]. In a cell-free system, glucosamine hydrochloride (0.05–0.8 mg/mL) potently scavenges superoxide anion in a concentration-dependent manner with more than 75% scavenging activity even at its lowest concentration [80,82]. At high concentrations, this compound inhibits deoxyribose oxidative damage by directly scavenging the hydroxyl radicals with approximately 55% inhibition at 3.2 mg/mL [80,82]. Glucosamine hydrochloride also has a concentration-dependent reducing power as measured by ferric-reducing antioxidant power assay [80,82]. However, it has a relatively weak ferrous ion-chelating effect [80,82]. A similar study by Mendis et al. revealed

that glucosamine hydrochloride and glucosamine sulphate (50–1000 µg/mL) could directly scavenge the superoxide anion, hydroxyl radical, and carbon-centred radicals as measured by electron spin resonance spectroscopy [71]. Furthermore, glucosamine sulphate shows greater antioxidant and radical scavenging activities than glucosamine hydrochloride at similar concentrations [71].

Glucosamine upregulates GSH (an antioxidant protein) and several antioxidant enzymes, including SOD, catalase (CAT), and glutathione peroxidase (GPx) in chondrocytes [71,83]. Glucosamine sulphate (10 mM) suppresses IL-1β-stimulated SOD2 upregulation in normal human articular chondrocytes probably due to the decrease in oxidative stress [83]. Glucosamine sulphate (1000 µg/mL), not glucosamine hydrochloride, remarkably upregulates the GSH level in SW1353 chondrocytes as early as 30 min [71]. Glucosamine hydrochloride (300 mg/kg/day oral intake for 60 days) can also increase the plasma GSH levels and the plasma SOD, CAT, and GPx activities in the chemical-induced rheumatoid arthritis in albino rats [75]. However, glucosamine sulphate (1 and 10 mM; 6 to 48 h) has upregulated the heme oxygenase-1 (HO-1) mRNA and protein level in primary human osteoarthritic chondrocytes, with or without IL-1β stimulation [84]. Reports on the anti-oxidative and anti-inflammatory actions of HO-1 [87–89] suggested that the induction of HO-1 expression serves as a negative feedback mechanism to protect cells against oxidative stress [90]. However, the exact role of HO-1 in OA is not conclusive.

Overproduction of ROS and reactive nitrogen species (RNS) is involved in OA pathogenesis [62,91,92]. The main ROS/RNS produced by chondrocytes are NO and superoxide anion which are converted into other free radicals including peroxynitrite, hydroxyl radicals, and $H_2O_2$, thus causing oxidative damages [93]. Endogenous superoxide anion is produced in the mitochondria, NADPH oxidase, and xanthine oxidase [91,94]. Valvason et al. stated that glucosamine sulfate (1 and 10 mM; 6 to 48 h treatment) could downregulate NAPDH oxidase subunit $p22^{Phox}$ level in primary human OA chondrocytes [84]. However, the effects of glucosamine on mitochondrial or xanthine oxidase-mediated ROS production in chondrocytes have not been investigated.

NO is specifically synthesized by NO synthase, including endothelial NOS, neuronal NOS, and iNOS [92,93]. Glucosamine hydrochloride could inhibit NO production in LPS-stimulated articular cartilage explant discs cultures (0.5–10 mg/mL, co-treatment with LPS) [85] and partly suppress NO production in primary chondrocytes isolated from non-OA patients (100 µg/mL; 24 h treatment) [86]. A 10 mM glucosamine sulphate treatment could completely abrogate IL-1β-stimulated NO production [84]. Glucosamine hydrochloride does not affect NO production on OA chondrocytes [86,95]. The molecular mechanism of glucosamine in suppressing NO production, particularly in non-OA and OA models, remains unknown. The only available study from Valvason et al. reported that glucosamine sulphate could completely suppress the IL-1β-stimulated iNOS upregulation [84]. Therefore, a decrease in NO production caused by glucosamine could be associated with its anti-inflammatory properties.

## 4. Activation of Autophagy by Glucosamine

Autophagy (self-digestion of cells) is a primary catabolic process wherein eukaryotic cells digest proteins or organelles inside the cytoplasm [17]. The content to be digested in the cytoplasm is surrounded by vesicles that combine with lysosomes to form autolysosomes. Macroautophagy, microautophagy, and chaperone-mediated autophagy are the three main types of autophagy [96]. Chondrocytes depend on autophagy as a reparatory mechanism during cellular damage to remove any damaged or dysfunctional organelles without compromising cartilage cellularity [17,97]. Rapamycin (an autophagy inducer) could improve the clinical manifestations of OA mice via enhancing autophagy [98].

Glucosamine induces autophagy in simian virus 40-transformed monkey kidney COS-7 cells [99], primary rat nucleus pulposus cells [100], human retinal pigment epithelial cells [101], human cervical adenocarcinoma Hela cells [99], and the nematode *Caenorhabditis elegans* [102]. However, the direct evidence of the role of glucosamine-induced autophagy in OA is scarce [103–105]. Glucosamine hydrochloride induces autophagy in primary chondrocytes derived from normal human articular cartilage [103,104] and hFOB1.19 osteoblasts [105]. Oral and intraperitoneal administration of

glucosamine (250 mg/kg body weight/day) on green fluorescent protein fused to light chain 3 (GFP-LC3)-transgenic mice for 7 days has induced autophagy in the liver and knee joint cartilage of mice, the magnitude of autophagy activation is greater than that of starvation [104]. Confocal microscopy analysis revealed that the chondrocytes present in the superficial and upper-middle zones of articular cartilage are more responsive to autophagy compared with those in the deep zone of cartilage [104]. Kang et al. reported that the high concentrations of glucosamine (50 and 100 mM) exhibit a biphasic effect on the activation of autophagy in primary healthy chondrocytes [103]. Short-term exposure (2 h) of glucosamine activates autophagy, whereas long-term exposure (24 h) shifts the autophagic response to apoptotic cell death [103]. High concentrations of glucosamine (> 10 mM) are cytotoxic to normal and OA chondrocytes [106,107]. Similarly, 24 h treatment of 100 mM glucosamine induces massive cytoplasmic vacuolation (a typical morphology of autophagic cell death) in primary normal rat chondrocytes [108]. Glucosamine at a concentration of 30 (6.5 mg/mL) and 116 mM (25 mg/mL) also induces prominent cell death in in vitro bovine cartilage explants with the typical cytoplasmic vacuolation [109]. The physiological effect of 4.34 μM glucosamine is observed upon the oral administration of a clinically relevant dose of glucosamine sulphate (1.5 g/day) [110]. Therefore, future research should focus on the physiologically relevant concentrations of glucosamine at different treatment time points.

The upstream initiation of glucosamine-mediated autophagy remains elusive. Current findings suggested that glucosamine-mediated chondrocyte autophagy may be associated with the protein kinase B (PKB or Akt)/Forkhead box O3 (FoxO3)/mammalian target of rapamycin (mTOR) pathway [104,105]. A remarkable and concurrent inhibition of Akt, FoxO3, and ribosomal protein S6 protein (a mTOR downstream target) phosphorylation is detected along with glucosamine-induced autophagy [104]. Similarly, glucosamine also induced hFOB1.19 osteoblast autophagy with the parallel suppression of mTOR phosphorylation [105] and triggered autophagy by activating 5' adenosine monophosphate-activated protein kinase (AMPK) and inhibiting mTOR pathway in primary rat nucleus pulposus cells and human retinal pigment epithelial cells [100,101]. However, glucosamine has induced mTOR-independent autophagy in COS-7 cells [99]. The role of Akt/FoxO3/mTOR/AMPK pathway in glucosamine-induced autophagy is not conclusive and thus requires further investigations.

Unfolded protein response (UPR) signaling is involved in the maturation of chondrocytes, osteoblasts, and OA pathogenesis [111]. During oxidative stress, UPR restores cellular homeostasis when an accumulation of unfolded/misfolded proteins occur which resulted from oxidative damages [111,112]. UPR signalling is reported to be associated with the endoplasmic reticulum (ER) stress-induced autophagy [112]. Glucosamine induces ER stress and GRP78 upregulation in the atherosclerosis models [113–117]. To the best of our knowledge, no study has investigated the role of glucosamine-induced ER stress or UPR signalling in OA. Only one relevant work by Calamia et al. reported that 2 h glucosamine sulphate (10 mM) treatment remarkably upregulated the UPR proteins, including binding immunoglobulin protein (GRP78), heat shock cognate 70 (HSP7C), and protein disulfide-isomerase 1/3 (PDIA1/3) in IL-1β-treated normal human articular chondrocytes [83]. Inducing ER stress could be beneficial in delaying the OA onset [118]. Moreover, OA chondrocytes respond well to ER stress [119]. Therefore, the UPR/ER stress/autophagy axis can be the next therapeutic target in glucosamine treatment.

## 5. Induction of Tissue Regeneration, Stem Cell Proliferation, and Differentiation by Glucosamine

Glucosamine and its derivatives inhibit osteoclastic differentiation [120] and enhance stem cells proliferation [121–124], chondrogenic differentiation or chondrogenesis [106,122–127] and osteoblastic differentiation [120]. Glucosamine (0.1 and 1 mM) and N-acetyl glucosamine induced differentiation of mouse MC3T3-E1 osteoblasts by increasing the mineralisation of extracellular matrix and the expression of osteopontin (middle-stage osteoblastic differentiation biomarker) and osteocalcin (late-stage osteoblastic differentiation biomarker) [120]. Meanwhile, early-stage osteoblastic differentiation biomarkers, including type I collagen and alkaline phosphatase, are not changed [120].

Glucosamine and N-acetyl glucosamine simultaneously suppressed the osteoclastic cell differentiation on MC3T3-E1 osteoblasts with the downregulation of the receptor activator of NF-κB ligand (RANKL) expression [120]. This observation was relevant to the protective effects of glucosamine in OA because impaired subchondral bone structure could weaken the ability of the joint to withstand mechanical loading, thus damaging the cartilage.

Derfoul et al. reported that glucosamine hydrochloride (100 μM and 1 mM for 11 days) promoted chondrogenesis in human mesenchymal stem cells (hMSCs) by upregulating the type II collagen, aggrecan, and sulfated GAG levels [106]. Glucosamine also maintained and restored the chondrogenic phenotypes in normal and OA chondrocytes in pellet cultures [106]. Additionally, glucosamine partially blocked IL-1β-mediated downregulation of type II collagen and aggrecan gene expression and inhibited the *MMP-13* gene expression in both normal and OA chondrocytes, as well as hMSCs [106]. Glucosamine treatment (2 mM for 21 days) in combination with glycoproteins, hyaluronic acid, chondroitin sulphate, and heparin sulphate (in chondrogenic differentiation media) can promote the chondrogenesis in murine embryonic stem cells (ESCs) encapsulated in a polyethylene-glycol (PEG)-based hydrogel (a 3D culture) [125]. Glucosamine treatment remarkably increased the size of ESC organoids with the accumulation of GAG; downregulation of type X collagen gene expression; and upregulation of aggrecan, type II collagen, and transcription factor Sox-9 gene expression [125]. Mechanistically, glucosamine stimulates ESC proliferation via the glucose transporter-2-dependent increase for glycolytic- and glutamine-derived intermediates, which could reduce the dependence on the hexosamine biosynthetic pathway [121].

Glucosamine (4.6 mM) also reduces calcium deposition and mineralisation in chondrogenic ATDC5 cells through the upregulation of sulphated GAG level by suppressing the expression of *C. elegans* small-worm-phenotype and Drosophila Mothers-Against-Decapentaplegic 2/4 (SMAD2/4) and matrix γ-carboxyglutamate protein (*MGP*) genes [128]. The glucosamine analogue, N-acryloylglucosamine, has similar properties in promoting ATDC5 cell proliferation and chondrogenesis but with lesser cytotoxicity [123]. The N-acryloylglucosamine-grafted PEG-based hydrogel also enhances the bone-derived hMSCs proliferation and chondrogenesis but inhibits the formation of fibrocartilage and hypertrophic chondrocytes [124]. The N-acyloylglucosamine-modified hydrogels also produce a better cartilage-like tissue block with abundance of proteoglycans and type II collagen secretion in the subcutaneous transplantation of NOD/SCID mice [124]. Similarly, 1% (*w/v*) glucosamine-incorporated silk fibroin:chitosan scaffold also promoted the umbilical cord blood-derived hMSCs proliferation and chondrogenesis in monolayer and spinner flask dynamic cultures [122].

Kamarul et al. investigated the effects of the combination of autologous chondrocyte implantation (ACI) surgical treatment with the oral treatment of glucosamine sulphate (120 mg/day) in male New Zealand white rabbits with surgical-induced knee joint cartilage damage [126]. Glucosamine remarkably improved the hyaline cartilage regeneration on ACI repair sites with the upregulation of proteoglycans, type II collagen and GAG expression compared with ACI alone [126]. The regeneration of these repair sites was remarkably improved with prolonged intake of glucosamine or in combination with chondroitin sulphate [126]. Therefore, glucosamine-mediated chondrogenesis could be a plausible mechanism for its cartilage regeneration effects. Further studies were required to investigate the role of the glucosamine-mediated chondrogenesis and stem cell proliferation in OA.

Table 1 summarizes the molecular mechanisms of glucosamine on OA.

**Table 1.** Molecular mechanisms of glucosamine on OA

| References | Study Design | Findings |
|---|---|---|
| **Anti-inflammation (↓ MMP levels)** | | |
| [33] | Osteosarcoma cell lines were treated with 10, 50, and 100 µg/mL glucosamine, and MMP-3 and MMP-9 were assessed | ↓ MMP-3 protein levels in cell lines |
| [129] | 186 different proteins secreted by chondrocytes from patients with osteoarthritis (OA) and drug treatment for 29 days | Glucosamine in combination with chondroitin was more effective in modulating the synthesis of proteoglycans and collagens than glucosamine alone. |
| [34] | Human chondrocytes were treated with glucosamine sulfate (0.1–10 mM). | ↑ the mRNA expression and protein levels of SIRT1 and its downstream gene *COL2A1* in chondrocyte ↓ MMP-1 and MMP-9 expression |
| **Anti-inflammation (↓ IL and TNF levels)** | | |
| [36] | In vitro and in vivo rat models of OA induced by ACLT were treated for 3 months with 175 mg/kg glucosamine sulfate | Protected against cartilage degradation↓ levels of inflammatory proteins in the affected knee ↓ IL-1β and TNF-α |
| [52] | Rats received glucosamine for 16 days orally at doses of 20, 40, 80, or 160 mg/kg/day | ↓ IL-6 and TNF levels ↑ serum nitrite |
| [53] | Mice with OA were treated for 15 days with glucosamine at doses of 40, 80, and 160 mg/kg/day Rats with OA were treated with 50 or 100 mg/kg glucosamine for 2 months | ↓ joint swelling and OA symptoms ↓ IL-β, IL-6, TNF-α, and IL-2 ↑ cytokine IL-10 and ↓TGF-1 levels |
| [55] | Wistar rats were treated for 1 month with 500 mg of glucosamine sulfate/kg with chondroitin sulfate | ↓ IL-1β in serum |
| [56] | Patients with temporomandibular joint OA treated for 1 month or 1 year with 40 and 80 mg of glucosamine with hyaluronic acid (vs placebo with hyaluronic acid) | ↓ IL-1β after one month of treatment ↑ TGF-β, ↓ IL-1β, and IL-6 after one year of treatment |
| **Antioxidant (direct scavenging of free radicals)** | | |
| [80,82] | A cell-free system with the production of superoxide anion and hydroxyl radicals | Scavenge superoxide anion radicals and hydroxyl radicals |
| [80,82] | Cell-free FRAP assay and ferrous ion-chelating assay | A concentration-dependent reducing power but weak ferrous ion-chelating activity |
| [71] | A cell-free system as measured by electron spin resonance spectroscopy | Directly scavenge the superoxide anion, hydroxyl radical, and carbon-centered radicals. Glucosamine sulfate is more potent than hydrochloride form |
| **Antioxidant (upregulation of antioxidant protein/enzymes levels)** | | |
| [83] | Normal human articular chondrocytes were treated with glucosamine sulfate (10 mM) | ↓ IL-1β-stimulated SOD2 upregulation |
| [71] | SW1353 chondrocytes were treated with glucosamine sulfate and hydrochloride (50–1000 µg/mL) | ↑ GSH level upon glucosamine sulfate treatment but not glucosamine hydrochloride |
| [84] | Primary human osteoarthritic chondrocytes were treated with glucosamine sulfate (1 and 10 mM) | ↑ *HO-1* mRNA and protein level |
| **Antioxidant (suppression of free radical production)** | | |
| [84] | Primary human osteoarthritic chondrocytes were treated with glucosamine sulfate (1 and 10 mM) | ↓ NADPH oxidase subunit p22$^{Phox}$ level |
| [85] | Articular cartilage explant discs cultures were co-incubated with LPS and glucosamine hydrochloride (0.5–10 mg/mL) | ↓ NO production |
| [86] | Primary non-OA chondrocytes were treated with glucosamine hydrochloride (100 µg/mL) | ↓ NO production (partially) |
| [84] | Primary human osteoarthritic chondrocytes were treated with glucosamine sulfate (1 and 10 mM) | ↓ IL-1β-stimulated NO production and iNOS upregulation |

**Table 1.** *Cont.*

| References | Study Design | Findings |
| --- | --- | --- |
| | Activation of the autophagy | |
| [103,104] | Human chondrocytes were treated with glucosamine (10–100 mM) for 2 and 24 h | High doses ↓cartilage degradation↓ peroxisomal oxidation |
| [104] | Human articular cartilage chondrocytes were treated with glucosamine (0.1–10 mM) for 8 and 24 h | ↑ LC3-II/LC3-I ratio ↑ autophagy flux Inhibited Akt/FoxO3/mTOR phosphorylation |
| | GFP-LC3-transgenic mice were treated with glucosamine (250 mg/kg body weight/day) via oral and intraperitoneal administration for 7 days | ↑ autophagy in the mice liver and knee joint cartilage |
| [105] | Human hFOB1.19 osteoblasts were treated with glucosamine (0.2–1 mM) up to 48 h | ↑ LC3 II and Beclin 1 ↓ SQSTM1/p62 Inhibited mTOR phosphorylation |
| [83] | Normal human articular chondrocytes were treated with glucosamine sulfate (10 mM) and then stimulated with IL-1β (10 ng/mL) | ↑ GRP78, HSP7C and PDIA1/3 protein levels |
| | Tissue regeneration (stem cell proliferation and differentiation) | |
| [121] | Low glucose-isolated ESCs that highly expressed glucose transporter 2 were treated with glucosamine (0.8 and 2 mM) for 4 days | ↑ ESCs proliferation |
| [106] | hMSCs were treated with glucosamine hydrochloride (100 μM and 1 mM) for 11 days | ↑ chondrogenesis (↑ type II collagen, aggrecan, and sulfated GAG levels) |
| [106] | Primary normal and OA chondrocytes were treated with glucosamine hydrochloride (100 μM and 1 mM) for 11 days | ↑ chondrogenic phenotypes with the restoration of type II collagen, aggrecan, and sulfated GAG levels Partially blocked IL-1β-mediated downregulation of type II collagen and aggrecan genes expression and inhibited the MMP-13 gene expression |
| [125] | 3D-culture of murine ESCs were treated with 2 mM glucosamine, together with glycoproteins, hyaluronic acid, chondroitin sulfate, and heparin sulfate for 21 days | ↑ ESC organoids size ↑ chondrogenesis (↑ GAG, aggrecan, type II collagen and Sox-9 genes expression and ↓ type X collagen) |
| [128] | Chondrogenic ATDC5 cells were treated with 4.6 mM glucosamine for 5 and 35 days | ↓ calcium deposition and mineralization with ↑ sulfated GAG level and ↓ *SMAD2/4* and *MGP* genes expression |
| [120] | MC3T3-E1 osteoblasts were treated with glucosamine (0.1 mM and 1 mM) and N-acetyl glucosamine for 3 and 21 days | ↑ osteoblastic differentiation (↑ osteopontin and osteocalcin levels) ↓ osteoclastic differentiation (↓ RANKL level) |
| [126] | Male New Zealand white rabbits with surgical-induced knee joint cartilage damage were received ACI surgical repair treatment with or without the oral gavage of glucosamine sulphate (120 mg/day) for 3 and 6 months | Glucosamine improved the hyaline cartilage regeneration on ACI repair sites with ↑ proteoglycans, type II collagen and GAG expression |

Abbreviations: ↑: increase or upregulate; ↓: decrease or downregulate; ACLT: anterior cruciate ligament transection; ACI: autologous chondrocyte implantation; Akt: protein kinase B; COL2A1: collagen type 2A1; ESCs: embryonic stem cells; FRAP: ferric reducing antioxidant power; FoxO3: Forkhead box O3; GAG: glycosaminoglycan; GRP78: binding immunoglobulin protein; GSH: glutathione; hMSCs: human mesenchymal stem cells; *HO-1*: heme oxygenase 1 gene; HSP7C: heat shock cognate 7; IL: interleukin; iNOS: inducible NO synthase; LC3-I: light chain 3 subunit I; LC3-II: light chain-subunit-II; LPS: lipopolysaccharide; *MGP*: matrix γ-carboxyglutamate gene; MMP: matrix metalloproteinase; mTOR: mammalian target of rapamycin; NADPH: nicotinamide adenine dinucleotide phosphate; NO: nitric oxide; RANKL: receptor activator of NF-κB ligand; OA: osteoarthritis; PDA1/3: protein disulfide-isomerase 1/3; *SMAD 2/4*: *Caenorhabditis elegans* small-worm-phenotype and Drosophila Mothers-Against-Decapentaplegic 2/4 gene; SOD2: superoxide dismutase 2; SQSTM1/p62: sequestosome 1; TGF: transforming growth factor; TNF: tumor necrosis factor.

## 6. Contradictory Effects of Glucosamine on Joint Health

Studies showed that glucosamine did not improve OA [130–136]. Roman-Blas et al. treated rabbits with OA with a combination of glucosamine sulphate (1500 mg) and chondroitin sulphate (1200 mg) daily for 2 weeks [130]. Similar proinflammatory profiles were observed for the rabbits with OA and the treated rabbits [130]. Additionally, magnetic resonance imaging showed that the joint structure of individuals with chronic knee pain did not change after a 24-week glucosamine supplementation (1500 mg per day) [131]. Supplementation of glucosamine (1500 mg per day) or chondroitin for 2 years also did not cause remarkable changes in the joint structure of OA patients (aged 45–75 years) [132]. However, the combination treatment of glucosamine and chondroitin remarkably reduced joint space narrowing [132]. Cahlin et al. studied the effects of glucosamine sulphate on 59 patients with temporomandibular OA [133]. These subjects were randomly divided into two groups, where the first group took 1200 mg of glucosamine sulphate daily, and the second group took a placebo [133]. After 6 weeks, the group receiving glucosamine treatment showed no difference in all signs and symptoms of OA compared with those of the placebo group [133]. Furthermore, several meta-analyses also showed either negligible [134] or small pain-relieving effects of glucosamine in OA patients [135,136]. The discrepancies in the effectiveness of glucosamine on OA between various studies should be further examined.

## 7. Conclusions

Glucosamine is an agent that has beneficial effects on the joint structure and maintains joint health by preventing the degradation of cartilage, reducing the inflammation and oxidative stress of the joint, improving the autophagy response of the chondrocytes, and increasing the chondrogenic potential of stem cells. Thus, glucosamine functions as a building block of the cartilage matrix and has multifaceted roles in promoting joint health. Further studies may enlighten additional functions of this interesting molecule. Glucosamine may also have a synergistic effect with other anti-osteoarthritic agents, but this speculation needs further investigation. Figure 1 summarizes the molecular mechanisms of glucosamine in protecting against osteoarthritis.

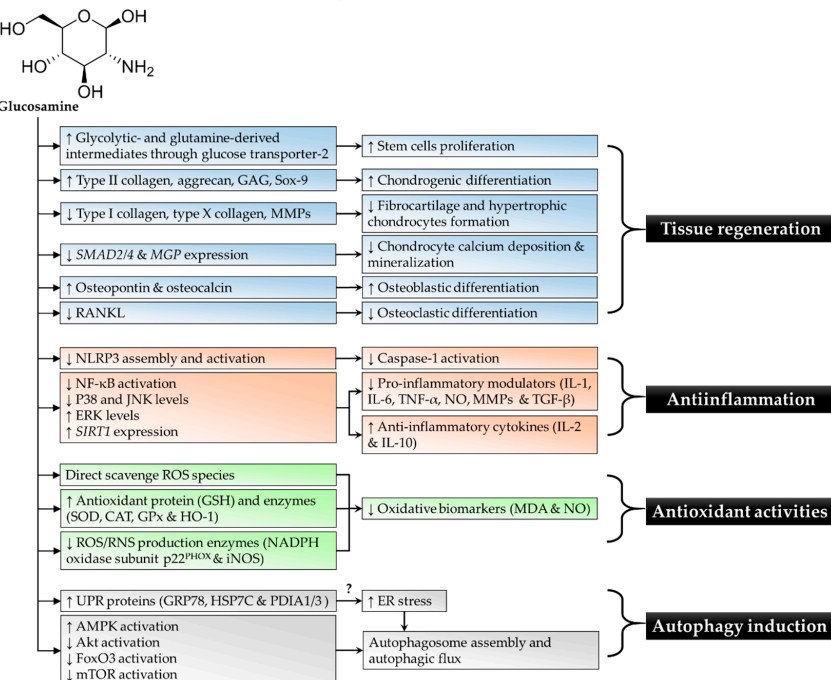

**Figure 1.** Summary of the molecular mechanisms of glucosamine in protecting against OA. Abbreviations: ↑: increase; ↓: decrease; ?: possible involvement; Akt: protein kinase B; AMPK: 5′ adenosine monophosphate-activated protein kinase; CAT: catalase; ER: endoplasmic reticulum; ERK:

extracellular-signal-regulated kinase; FoxO3: Forkhead box O3; GAG: glycosaminoglycan; GRP78: binding immunoglobulin protein; GPx: glutathione peroxidase; GSH: glutathione; HO-1: heme oxygenase 1; HSP7C: heat shock cognate 7; IL: interleukin; iNOS: inducible NO synthase; JNK: c-Jun N-terminal kinase; MDA: malondialdehyde; MGP: matrix γ-carboxyglutamate protein; MMPs: matrix metalloproteinases; mTOR: mammalian target of rapamycin; NADPH: nicotinamide adenine dinucleotide phosphate; NF-κB: nuclear factor κB; NLRP3: nucleotide-binding oligomerization domain-like receptor containing pyrin domain 3; NO: nitric oxide; PDA1/3: protein disulfide-isomerase 1/3; RANKL: receptor activator of NF-κB ligand; RNS: reactive nitrogen species; ROS: reactive oxygen species; SIRT1: sirtuin-1; SMAD2/4: *Caenorhabditis elegans* small-worm-phenotype and *Drosophila* Mothers-Against-Decapentaplegic 2/4; SOD: superoxide dismutase; TGF-β: transforming growth factor- β; TNF-α: tumor necrosis factor-α.

**Author Contributions:** Conceptualization, H.M.A.-S. and K.-Y.C.; Literature search, H.M.A.-S. and K.-L.P.; Writing—original draft preparation, H.M.A.-S., K.-L.P., and K.-Y.C.; Writing—review and editing, K.-L.P., S.I.-N., and K.-Y.C.; Supervision, S.I.-N. and K.-Y.C.; Funding acquisition, K.-Y.C.

**Funding:** The researchers are funded by grants FRGS/1/2018/SKK10/UKM/03/1 provided by the Ministry of Education, Malaysia and FF-2018-405 provided by Universiti Kebangsaan Malaysia.

**Acknowledgments:** We thank the Ministry of Education, Malaysia (FRGS/1/2018/SKK10/UKM/03/1) and Universiti Kebangsaan Malaysia (FF-2018-405) for funding this project.

**Conflicts of Interest:** The authors declare no conflict of interest.

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
