# Peer review of "Multifaceted Protective Role of Glucosamine against Osteoarthritis: Review of Its Molecular Mechanisms"

_scipharm, doi:10.3390/scipharm87040034_

Round 1

Reviewer 1 Report

This review offers a revision of the literature on the mechanisms by which glucosamine would prevent cartilage decay during osteoarthritis.  The evidence suggests that beyond its role as precursor for building cartilage matrix, glucosamine can act as anti-inflammatory, antioxidant and stimulates autophagy in chondrocytes.   I have the following comments:

The bullet number 3 in line 241. “3. Activation of autophagy by glucosamine” should be 4 and the subsequent numbers need to be increased by one. The reference [29] in line 64 does not support the strong authors’ statement “The ability of glucosamine-containing nutraceuticals to reduce proteoglycan loss, impede cartilage degeneration, delay joint-space narrowing and improve pain has been widely reported [29]” The paragraph on the mechanism of glucosamine stimulating autophagy is not clear.  The authors do not provide enough information to understand how glucosamine increases authophagy until it reaches  certain dose and then decreases.  Does it mean glucosamine is only beneficial within small window of concentrations? What concentration range is expected in vivo in the joint? A graphic diagram connecting the different intracellular events associated with induction/inhibition of autophagy and protection against OA should help facilitating the description of this section. The title should include “osteoarthritis” since the authors do not talk about protecting joint health against RA or gout.

Author Response

Dear Reviewer, 

Thank you for your constructive comments. We have responded to your comments in the attached response sheet. We hope you find the replies satisfactory. 

Thank you. 

Reviewer 2 Report

The manuscript is an overview of the studies on anti-inflammatory, antioxidant and autophagy inducing effects of glucosamine that are involved in the mechanism of action of glucosamine in preventing osteoarthritis. The manuscript has reviewed the current literature and included all major important articles published in this area. However, considering that osteoarthritis is a degenerative disease and manifests an insufficient regeneration of the cartilage my suggestion will be to include an additional section summarizing the effects of glucosamine on tissue regeneration and emphasizing the need for future studies to delineate the gucosamine effects on various cell types and sub-populations of stem cells involved in chondrogenesis.

Minor comments:

The manuscript requires language editing for improving the readability and to correct minor grammatical errors.

Author Response

Dear Reviewer, 

Thank you for your constructive comments. We have responded to your comments in the attached response sheet. We hope you find the replies satisfactory. 

Thank you. 

This manuscript is a resubmission of an earlier submission. The following is a list of the peer review reports and author responses from that submission.